# A Case Study in Citizen Science: The Effectiveness of a Trap-Neuter-Return Program in a Chicago Neighborhood

**DOI:** 10.3390/ani8010014

**Published:** 2018-01-18

**Authors:** Daniel D. Spehar, Peter J. Wolf

**Affiliations:** 1Independent Researcher, 4758 Ridge Road, #409, Cleveland, OH 44144, USA; danspehar9@gmail.com; 2Best Friends Animal Society, 5001 Angel Canyon Road, Kanab, UT 84741, USA

**Keywords:** trap-neuter-return, TNR, free-roaming cats, feral cats, stray cats, sterilization, management, citizen science

## Abstract

**Simple Summary:**

Strong public support in the United States for the non-lethal management of free-roaming cats has prompted an increase in the practice of trap-neuter-return (TNR) over the past quarter-century, yet a paucity of analyzable data exists. Data sets collected by citizen scientists are likely to play an important role in filling this information void. A citizen scientist in Chicago, Illinois, recorded significant reductions in a free-roaming cat population as the result of a neighborhood TNR program. Colony populations, when grouped by the number of years enrolled in the program, declined by a mean of 54% from entry and 82% from peak levels. Results from concurrent TNR programs in the Chicago area are compatible with these findings.

**Abstract:**

The use of trap-neuter-return (TNR) as a method of managing free-roaming cat populations has increased in the United States in recent decades. Historically, TNR has been conducted most often at a grassroots level, which has led to inconsistent data collection and assessment practices. Consequently, a paucity of analyzable data exists. An initiative is underway to standardize TNR program data collection and assessment. However, it could be some time before scientifically sound protocols are implemented on a broad scale. In the interim, sets of data collected by nascent citizen scientists offer valid opportunities to evaluate grassroots TNR programs. The purpose of the present study was to examine the effectiveness of a TNR program conducted by a citizen scientist located in Chicago, Illinois, where a county law permitting TNR was enacted in 2007. Colony populations, when grouped by the number of years enrolled in the program, declined by a mean of 54% from entry and 82% from peak levels. Results from coexistent TNR programs in the Chicago area are consistent with these findings.

## 1. Introduction

Strong public support for the non-lethal management of free-roaming cats [1,2,3,4] has led to an increase over the past 25 years in the practice of trap-neuter-return (TNR) [5,6]. As recently as a decade ago, the vast majority of TNR efforts were conducted by individuals and small groups. Since that time, TNR has been assimilated into the best practices of mainstream animal welfare organizations [7,8,9,10]. Advocates of TNR claim that it reduces free-roaming cat populations over time by preventing the birth of additional cats [5,11]. However, critics assert that TNR programs fail to produce convincing evidence of diminished free-roaming cat populations [12,13,14].

Various studies have documented declines in free-roaming cat populations due to TNR [15,16,17,18], including the elimination of individual colonies [19,20] and reduction [21] or elimination [22] of kitten births. Yet, it has not been typical practice for TNR programs to systematically document and assess their results [6,23]. Concerns about the resultant lack of analyzable data have prompted a movement toward adoption of standardized recordkeeping and data assessment practices [6,23,24].

Notable exceptions to the typical level of documentation associated with TNR efforts are programs conducted by nascent citizen scientists, who often take it upon themselves to collect applicable data sets and record their results. Citizen science has been embraced by various conservation organizations as a source of valuable information; it has been cited in peer-reviewed scientific literature and has directly influenced action outcomes [25]. One such example is the National Audubon Society’s 118-year-old Christmas Bird Count, which the organization promotes as “the nation’s longest-running citizen science bird project” [26]. Until such time that typical TNR program data collection and assessment practices have become standardized, information recorded by fastidious citizen scientists will likely play an important role in evaluating the effectiveness of TNR at reducing free-roaming cat populations. 

The purpose of the present study was to examine the effectiveness of a TNR program, known as Cats In My Yard (CIMY), conducted by a citizen scientist in Chicago, Illinois, the third largest city in the US with a population of 2.7 million people [27].

The CIMY program began in 2004 as an attempt to catch a single ill cat, but within three years had evolved into a systematic initiative to trap, sterilize, and return or adopt all the free-roaming cats in an entire neighborhood of Northwest Chicago (Figure 1) [28]. The CIMY TNR program presented a unique opportunity to evaluate colony-level and neighborhood-level population data collected within a broader setting, Cook County, where detailed data sets (though not without their own shortcomings) were available to provide additional context.

## 2. Materials and Methods

### 2.1. Program Origins and Design, and Site Description

Initially, CIMY founder Vanessa Smetkowski trapped, sterilized, vaccinated, and returned or transferred to rescue groups for adoption only those free-roaming cats who frequented her yard [28]. Soon after the Managed Care of Feral Cats (MCFC) ordinance, a Cook County policy authorizing the formal establishment and management of feral cat colonies was implemented in 2007, however, she began working with other residents of her Humboldt Park neighborhood to sterilize and return cats on their properties as well. At the same time, Smetkowski began to diligently track the cats who appeared in her yard, as well as at the growing number of sites near her home where she assisted others [29]. Smetkowski provided all the raw data for the CIMY colonies referenced in this paper (see Appendix A).

The MCFC ordinance allows for resident volunteers, under the supervision of sponsoring humane societies, to register and manage colonies of free-roaming cats. Management includes trapping, sterilizing and vaccinating cats (against rabies, at a minimum) via licensed veterinarians, and either returning them to the location of capture or arranging for adoption, as well as for providing ongoing care [30]. In addition, the MCFC requires microchipping of cats trapped as part of the program [31]. Smetkowski maintains that the mandatory microchipping provision of the ordinance was her primary motivation for closely tracking outcomes for the cats she handled. All cats trapped as part of the CIMY program were microchipped [29]. Smetkowski did not possess a microchip scanner, so she did not track outcomes for the cats via microchip. However, she was contacted by various entities about eight different cats who were microchipped as part of the program. 

Smetkowski’s urban Chicago neighborhood covers an area of 9.3 km^2^ and is predominantly made up of single- and two-family homes, typically on 7.6 × 38.1 m lots, and small apartment buildings [32]. As of the 2010 census, the total human population of the Humboldt Park community was 56,320 [33]. The eastern portion of Humboldt Park, where the TNR program described here took place, has experienced rapid gentrification in recent years [34], resulting in the displacement of a significant number of cats from buildings targeted for demolition or renovation [32].

Between 2007 and 2016, CIMY trapped, sterilized, vaccinated, and returned or adopted free-roaming cats at 20 discrete colony sites within an estimated one-square-kilometer area near Smetkowski’s home [35]. Smetkowski assigned names to all of the colonies and to the vast majority of individual cats. She kept track of each cat’s color and markings, sex, age group (adult or kitten up to 12 weeks of age), perceived level of socialization (“feral” or candidate for adoption), and spay-neuter, vaccination, ear-tip, and microchip status, as well as post-surgery outcome (i.e., returned to colony or admitted to a sponsoring humane society for adoption). New arrivals and colony member deaths and disappearances (defined as the failure to observe a particular cat, without knowledge of his or her whereabouts, for six months or more) also were recorded [32].

Smetkowski trapped and returned the vast majority of the cats herself, while other colony feeders and caretakers performed the balance of the work involved. Smetkowski located colonies by knocking on doors in order to collect information from neighbors, as well as by following easily observable free-roaming cats back to their respective sources of food [29]. She supplied colony caretakers with insulated plastic storage bins to act as shelters for the cats, as needed [32]. Feeding stations were set up in caretakers’ yards or in nearby alleyways.

Sterilization surgeries, rabies and feline viral rhinotracheitis/calicivirus/panleukopenia (FVRCP) vaccinations, ear-tipping, and microchipping were performed at local low-cost clinics and funded either out-of-pocket or by donations and grants. The cost for these services, in addition to flea and ear mite treatment when warranted, ranged from $26.00 to $38.00 per cat, depending upon the facility used. Cats were euthanized only for serious health reasons, and testing for feline immunodeficiency virus (FIV) and feline leukemia (FeLV) was done only when cats appeared to be sick or when trapped cats were determined to be suitable candidates for adoption [32].

Six Cook County humane societies served as sponsoring organizations for registered feral cat colonies in 2016 [36]. In accordance with provisions of the MCFC, these organizations are responsible for collecting data from colony caretakers and reporting their findings annually to the Cook County Department of Animal and Rabies Control (CCDARC), conducting public education campaigns, and mitigating issues that arise concerning registered colonies [37].

### 2.2. Data Collection

Details regarding cat description and location were recorded by Smetkowski on paper as each cat was trapped, and stored, along with a photo (when available) in a folder designed for the particular colony. Records were updated with additional information (e.g., sex) and transferred to a spreadsheet once each cat was returned to the appropriate colony or transferred to a sponsor organization.

Smetkowski updated census information for each colony on an annual basis, at minimum, through a combination of visual observation (year-round) and informal interviews with colony caretakers (in December of each year). However, it was not unusual for her to track changes in colony membership more frequently. Smetkowski regularly jogged the streets of her neighborhood, past many of the colony sites, and often stopped to check on the cats and their caretakers. She also placed calls to caretakers for updates when she failed to encounter them for more than a few months on her trips through the neighborhood. In addition, Smetkowski would collect census information when colony caretakers contacted her for assistance. 

For the purposes of this study, CIMY spreadsheets were transferred to Microsoft Excel for analysis. Information (e.g., shelter statistics or TNR program data) from other Cook County sources, including Chicago Animal Care and Control (CACC), CCDARC, and Tree House Humane Society (THHS) were collected and examined in order to provide context regarding what was occurring concurrently in the community at large. Unstructured interviews were conducted with Smetkowski and key representatives of CACC, CCDARC, and THHS to clarify details and fill information gaps. Descriptive statistics were calculated for key population variables.

## 3. Results

### 3.1. Cats in My Yard

Over a ten-year period, Smetkowski trapped, sterilized, vaccinated, and returned or adopted out 195 cats from the 20 existing colonies near her home. The program formally began in 2007 with the tracking of cats in Smetkowski’s own back yard. Other colonies entered the program intermittently, as they were discovered by Smetkowski over six subsequent years. Specifically, a second existing colony near her home was added in 2008, followed by three more in 2009, six in 2010, one in 2011, six in 2012, and two in 2013. In aggregate, 75 cats were recorded at colony entry (Table 1). Additional cats were contemporaneously reported by caretakers at a number of these locations, but such cats were not counted as part of the program until they were subsequently trapped for sterilization or, at minimum, observed and recorded by Smetkowski herself. According to Smetkowski, unrecorded cats already on site likely accounted for the majority of what appear to be spikes in population occurring in a number of colonies up to three or four years after entry (Table 1) into the program [38]. Furthermore, Smetkowski suggests that undocumented movement of cats among several of the colonies may have also contributed to this phenomenon, but likely to a lesser extent [39]. Regardless of the cause, fluctuations in free-roaming cat colony size are not uncommon [19]; similar variations are oftentimes observed among localized wildlife populations [17]. Only four cats were known to have emigrated from one colony to another within the 20-colony CIMY program area during the study period.

Adult cats trapped as part of the program outnumbered kittens 176 (90.3%) to 19 (9.7%). Of cats whose sex was identified (159/195), 62.3% were male and 37.7% female. The sex of 36 cats was unknown to Smetkowski because, unlike the others, they were not sterilized immediately after being trapped. Of these cats, 13 were kittens transferred directly to rescue because they did not yet meet the minimum age and weight requirements for sterilization surgery, while the remainder were returned to their owners or found to have an existing ear-tip, signifying that they had been sterilized as part of previous TNR efforts. In total, 180 (92.3%) of the cats in the program were sterilized; 155 (79.5%) of the total were spayed or neutered after being trapped, while 25 (12.8%) had previously been sterilized (Table 2). 

At year-end 2016, 44 (22.6%) of the 195 total cats trapped as part of the program remained on site. The maximum number of cats in each of the 20 colonies ranged from one to 35 (median of 5.5). The greatest total population across colony sites occurred in 2012 when 88 cats simultaneously resided in 18 colonies. This number decreased each subsequent year until only 44 cats (across 20 colonies) were present at year-end 2016 (a reduction of 50%) (Table 3). When the total of each colony’s peak size (152), regardless of when the colony was “enrolled” into the program, is compared to the year-end 2016 census, a total reduction of 71.1% occurred (Table 4). Grouping the 20 colonies according to the number of years for which detailed tracking was conducted, reveals a mean population reduction of 54% (SD = 58.4%; median = 82%) from entry levels, and 82% (SD = 19.5%; median = 89%) from peak levels (Table 5).

In all, eight colonies were entirely eliminated; all but three colonies experienced reductions in size from peak levels (the others experienced no overall change), and fifteen colonies dropped in size from initial levels (11) or had the same population at the study’s end as at colony entry (4). In the five colonies that experienced an increase in size from entry levels, the total population rose from 18 to 31 cats. At entry, individual colonies ranged in size from one to nine (median of 3.0) cats. By the end of the study period, overall median colony size had been reduced to 1.0; of the twelve colonies not reduced to zero, colony size ranged from one to twelve cats (median of 3.0).

The last known litter of kittens produced by a cat living outdoors in the program area was born in 2009. In 2014, one additional litter of kittens was known to have been born outside when a group of seven cats was displaced from a home that was abruptly vacated and demolished. In this case, the mother cat and her three kittens were admitted to a local humane society for adoption, while another adult cat was removed by CACC, and two more adult cats were trapped, sterilized, and vaccinated before being relocated to an existing managed colony in a neighboring community [40].

Of the 195 total cats who were a part of the program between 2007 and 2016, 59 (30.3%) were adopted or admitted to a rescue organization for adoption, 67 (34.4%) disappeared, 6 (3.1%) were euthanized due to serious injury or illness, 13 (6.7%) died from others causes, 3 (1.5%) were returned to their owners, 2 (1.0%) were relocated outside of the program area, and 1 (0.5%) was seized by CACC. As described above, due to the presence of a microchip, Smekowski was contacted by various entities about eight of the cats: three times by CACC (two cats adopted and one returned to colony), twice by the Chicago Animal Welfare League (one cat adopted and one returned to colony), twice by private veterinary practices (both cats adopted), and once by MedVet emergency clinic (the cat was euthanized after being struck by a vehicle). Of the 44 cats who remained at colony sites, four are known to have left their original colony to take up residence in another colony within the program area.

### 3.2. Data from Other Cook County Sources 

#### 3.2.1. Tree House Humane Society

Sponsor-agency THHS utilized two separate grants from PetSmart Charities, Inc. to fund consecutive two-year targeted TNR programs in the city of Chicago [41]. The purpose of the first program, which ran from July 2011 to June 2013, was to sterilize and vaccinate free-roaming cats in two zip codes (60647 and 60651) [42]. The CIMY TNR project also was conducted in zip code 60647, but was done so independently of the THHS initiative. 

At the end of the first targeted program, 1500 free-roaming cats from the two designated zip codes had been trapped, sterilized, and vaccinated; 1155 (77%) were returned to locations where they were captured and 345 (23%) were removed by THHS for adoption [42]. Contemporaneous reports indicate that over the two-year program period CACC observed a 40% reduction in stray cat intake from zip code 60647 and a 30% decline from zip code 60651 [42]. 

The second THHS targeted TNR program, which was expanded to include four Chicago zip codes (60647, 60651, 60623, and 60624) and ran from July 2013 to June 2015, resulted in the sterilization and vaccination of 2000 free-roaming cats [43]. Of the total number of cats trapped, 1574 (78.7%) were returned to their respective colonies, while 426 (21.3%) were sociable enough to be adopted into homes. It was reported at the time that “more than [a] 50%” reduction in stray cat intake at CACC took place in the two carry-over zip codes (60647 and 60651) during the four-year combined program period [43] (p. 2). 

Attempts to obtain the raw zip code data from CACC required to corroborate these reported results were only partially successful [44,45,46,47]. Unfortunately, data from 2010, the baseline year used to measure results, is missing; however, data was acquired for 2009 and 2011 [48]. When results from the end of the first two-year program period were compared to the average of data points reported for 2009 and 2011 (as a substitute for the missing 2010 data), the reduction in stray cat intake for each zip code was greater than reported above: 84.7% for 60647 and 51.7% for 60651. Similarly, the reductions in stray cat intake for the four-year combined program period also were larger than reported when compared to the substitute baseline: 78.6% for 60647 and 62.7% for 60651 (70.2% combined). In addition, anecdotal reports indicated that all city aldermen representing districts within the targeted zip codes received fewer complaint calls related to outdoor cats during the study period; however, such information was not specifically tracked as part of the program [43].

#### 3.2.2. Chicago Animal Care & Control and Cook County

City-wide, CACC experienced a 26% reduction in stray cat intake from 2010 to 2015 [49]. For the purpose of comparison, it is estimated by the American Society for the Prevention of Cruelty to Animals that total feline intake at U.S. shelters decreased from 3.3 million to 3.2 million from 2011 to 2016, a decline of only 3% [50]. When 2016 CACC data [51] are compared to those of 2006 [52], the year prior to enactment of the MCFC ordinance, total feline intake (no breakdown between stray cats and owner-surrendered cats was available) had decreased by 53.9%.

From 2007, the year the MCFC ordinance was enacted, to year-end 2016 (with current results from all but one sponsoring organization reported), a total of 2381 feral cat colonies were registered in Cook County; colony sponsors reported 16,371 cats residing in those colonies (mean colony size of 6.9). At year-end 2016, a total of 21,312 cats had been sterilized under the purview of the MCFC ordinance, while 6410 free-roaming cats (30.1%) had been adopted into homes [36].

## 4. Discussion

Prior to enactment of the MCFC ordinance in 2007, free-roaming cats in the city of Chicago—as well as in municipalities throughout Cook County—were routinely trapped and euthanized in response to complaints from residents [53,54]. In fact, in 2006, CACC admitted 11,461 cats and euthanized 8588 (74.9%) of them [52], and it was not uncommon for smaller municipalities within the county to each trap and euthanize up to 500 free-roaming cats per year [55]. However, in 2016, intake at CACC had decreased to 5278 cats (3432 of them stray), and of that number only 785 (14.9%) were euthanized [51]. 

As was observed in Alachua County, Florida [56], it is likely that intense TNR activity in the city of Chicago contributed significantly to the reductions in feline intake and euthanasia at CACC. Intake was likely lowered because of two interrelated components of TNR: sterilization and adoption. Sterilization likely reduced intake due to fewer kitten births and a reduction in the number of impoundments initiated as a result of resident complaints concerning nuisance feline behaviors commonly associated with mating. Adoption allowed for the removal of sociable cats and kittens directly from TNR programs or after transfer to local rescue organizations, thus eliminating the need for admission to the city shelter. More than 30% of cats, both in the 20 CIMY colonies and across the 2381 colonies registered with Cook County post-2007, were either adopted directly from colonies or transferred to rescue groups for adoption. CACC was not a registered sponsoring organization under the MCFC, but cooperated with such groups so that impounded ear-tipped (and microchipped) cats were returned to their respective colonies; these cats were included in shelter intake data [57]. The euthanasia rate at CACC was likely reduced because lower feline intake lessened overcrowding, which in turn decreased the incidence of illness among impounded cats awaiting adoption [57]. The total number of free-roaming cats living in Chicago (and Cook County) is unknown. Therefore, trends in overall population could not be determined. Thus, the surrogate indicators of shelter cat intake and shelter cat euthanasia were examined to better understand the possible impact of TNR on population size at the community level, as has been done elsewhere [56,58]. Significant city-wide reductions at CACC in both feline intake and euthanasia appear to be associated with an increase in the use of TNR after the enactment of the MCFC ordinance in 2007. Moreover, intensive TNR efforts conducted by THHS in several targeted Chicago zip codes appear to be associated with even greater reductions in the same metrics. The influence of confounding variables (e.g., modifications to shelter policies, animal control policies, or changes in community attitudes) on the apparent associations between TNR and reductions in feline intake and euthanasia at CACC cannot be dismissed entirely, although no such factors were identified as part of this investigation.

Assessment of the CIMY program’s effectiveness at reducing a population of free-roaming cats was based upon direct outcomes and not reliant upon surrogate indicators. When colonies are categorized by the number of years they were tracked in the program, mean population declines of 54% from entry and 82% from peak levels took place (Table 5). These results are, broadly speaking, consistent with population modeling of sterilization programs implemented in “large urban areas”, where at least “15–20% of the reproductively active population is sterilized, every six months, on a sustained basis” [24] (p. 9). Again, entry size at several colony sites may have been underreported due to an idiosyncratic counting protocol whereby Smetkowski herself confirmed the presence of all cats before their inclusion in the program. The average point at which peak colony size was reached was between year two and year three after program entry (Table 1), which is consistent with the findings of Natoli et al. [16] and Nutter [19], who observed that it can take as long as three or more years before reductions in colony populations commence. 

The adoption of sociable cats and socializable kittens, commonly considered an essential part of TNR standard operating procedure [7,8,18], contributed significantly to the reductions in population size realized by the CIMY program. The percentage of cats put up for adoption from CIMY colonies (30.3%) was nearly identical to what occurred over the same time-period in Cook County as a whole (30.1%); the percentage of adoptions was smaller than at two TNR program sites in Florida [15,56], but similar to the estimated number of cats adopted from a TNR initiative in Newburyport, Massachusetts [20].

Disappearances (34.4%) exceeded adoption by a small margin as the most common recorded outcome for resident colony cats. The specific manner of disposition for cats in this category was unknown. Likely outcomes for missing colony cats include dispersal, death by traumatic injury (e.g., vehicle collision), abduction, and undeclared adoption by local residents [59]. Survey data indicates that 27% of pet cats in the US are acquired as strays [60]; thus, some cats who disappeared were likely adopted directly into homes. Undocumented migration between program colony sites was another possibility; if this occurred, according to Smetkowski it likely took place among three colonies (Martino Awesome, Ricky Martino, and V), each where the colony caretaker was periodically unable to dutifully monitor and report colony activity [39]. Such migrations may also help to account for the higher percentage of disappearances experienced by the CIMY program than observed at sites studied by Nutter (27%) [19] and Levy et al. (15%) [15].

Although Humboldt Park, an urban green space that shares its name with the neighboring community, is in close proximity to the CIMY colony sites (estimated distance ranged from 0.2 km to 1.2 km), coyote predation on the missing cats likely accounted for few, if any, of the disappearances. Gehrt et al. [59] concluded that cats and coyotes partition their respective use of urban habitats, with cats limiting their use of natural fragments and coyotes generally avoiding developed areas such as residences. Moreover, Smetkowski was unaware of any cat deaths caused by coyotes [61]. The free-roaming cats involved in the Gehrt [59] study, which was conducted in the Chicago metropolitan area, were found to be in good general health, which is consistent with findings from Alachua County, Florida [56] and San Jose, California [58], as well as the low incidence of euthanasia due to serious health concerns seen in the CIMY program.

A comparison of CIMY colony composition to what has been observed elsewhere produced mixed results. Of the cats for whom sex was identified, a similar ratio (1.6:1) of male to female cats was observed in two New York City neighborhoods as part of a TNR/population study conducted by Kilgour et al. [62]. Additional research has indicated that males have outnumbered females in free-roaming cat populations [63]; however, the opposite has also been found [64]. In 11 colonies monitored by Nutter [19], 51.5% of 344 total cats were male; nevertheless, females outnumbered males in all but one colony. Moreover, the seemingly low percentage of kittens (9.7%) trapped as part of the CIMY program was likely more a function of definition (kitten ≤ 12 weeks of age) than other factors. Studies reporting higher percentages of kittens, at times more than half of the cats trapped, assigned cats of up to six months of age to that category [15,56]. An atypically higher percentage of previously-neutered cats (12.8%) were trapped as part of the CIMY program, compared to the range of 2.3% to 5% that has been noted elsewhere [15,56,65]; however, this phenomenon was likely the result of widespread TNR efforts in adjacent communities.

## 5. Study Limitations

Limitations of the present study include those inherent in the use of historical records for research objectives, as has been noted elsewhere [66]. The availability of only incomplete or truncated sets of data documenting feline intake at CACC from specific zip codes is illustrative of this impediment. The limited scope of data collected and maintained regarding Cook County’s registered feral cat colonies is another example. Further research would be needed to thoroughly examine the associations between enactment of the MCFC ordinance in 2007, and the resultant widespread use of TNR in Cook County, and declines in feline intake at CACC. The impact of TNR on the county’s total free-roaming cat population might also warrant further investigation. 

It is possible that the idiosyncratic protocol employed by Smetkowski to count and document cats upon their enrollment into the CIMY program resulted in the underreporting of colony size at entry in several instances. As described above, this likely accounts for increases in reported colony size in affected colonies up to several years after program entry, as cats already present but not included in initial counts were officially documented and enrolled. Nevertheless, Smetkowski’s fastidious method of enrolling cats into the program likely prevented counts at colony sites from being overstated by colony caretakers via double-counting, as was described by Centonze and Levy [11].

Moreover, the CIMY TNR program was designed without control colonies to serve as benchmarks against which the impacts of program interventions could be measured. Such control groups, as established by Nutter [19], offer obvious benefits to researchers but are unlikely to be integrated into TNR efforts undertaken by citizen scientists whose primary interest is in reducing colony size as quickly as possible.

## 6. Conclusions

Much potential exists for citizen science to facilitate a better understanding of the impact of TNR on free-roaming cat populations. The CIMY case study outlined here fulfills this expectation. Nonetheless, it is important to consider that this TNR program began not as a scientific endeavor, but as a conscientiously executed, practical solution to a neighborhood dilemma: what to do about an abundance of free-roaming cats. Due to the diligence and fastidiousness of the TNR practitioner involved, mere do-it-yourself problem solving was elevated to a case of citizen science.

Nevertheless, as stated above, limitations to the generalizability of results are acknowledged and additional research is necessary to confirm the transferability of study findings. This qualification is somewhat tempered, however, by the considerable supplemental evidence presented to corroborate the CIMY results. A wide array of coexistent TNR activity in Cook County enabled multiple data sets to be examined. Although some data were missing or incomplete, the available evidence was consistent with CIMY findings.

Appeals for the professionalization and standardization of TNR data collection and assessment practices [23], both at the institutional and citizen-practitioner level, are warranted. More consistent and reliable data compilation processes across TNR program sites are needed to alleviate potential uncertainties concerning the validity of findings. Despite these limitations, however, results of the CIMY investigation appear to support previous findings [15,17,20] that TNR is capable of reducing free-roaming cat populations.

## Figures and Tables

**Figure 1 animals-08-00014-f001:**
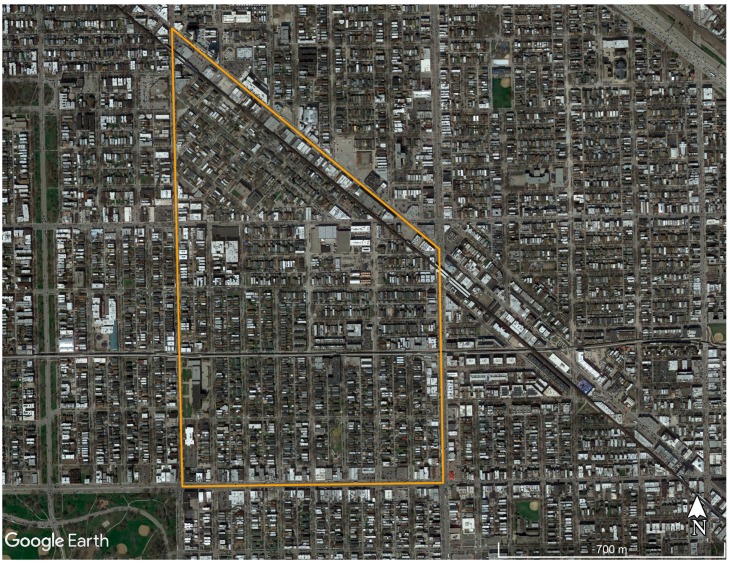
Boundaries of the Cats in My Yard trap-neuter-return (TNR) program area located in the Humboldt Park section of Chicago, IL, USA.

**Table 1 animals-08-00014-t001:** Colony size at entry, year-end follow-ups, and at peak.

Colony	Colony Population (Cats)											At Peak	
Name	Entry	YE1	YE2	YE3	YE4	YE5	YE6	YE7	YE8	YE9	YE10	Cats	Year
James Gang	3	1	1	3	4	6	5	6	5	6	5	10	3
Eleanor Rigby	4	4	8	7	7	4	2	1	0	0	-	16	2
Frontier	8	8	7	7	6	2	2	2	0	-	-	10	4
Garage Band	1	0	0	1	0	0	0	1	1	-	-	4	4
Martino Awesome	2	0	5	5	0	0	0	0	1	-	-	5	2
Jose Pussycats	3	3	3	8	6	6	5	5	-	-	-	11	3
Little Sister	4	0	3	6	6	6	6	6	-	-	-	6	3
Marta Volta	3	2	2	0	0	3	3	3	-	-	-	3	1
Mother	6	2	3	3	1	1	1	1	-	-	-	6	1
Ricky Martino	6	6	6	6	6	6	0	0	-	-	-	6	1
V	7	7	15	33	15	15	14	12	-	-	-	35	3
Bonita	3	3	0	0	0	0	0	-	-	-	-	3	1
La Vida Lydia	1	1	2	2	1	1	-	-	-	-	-	2	2
Major Tomcat	4	4	4	4	4	4	-	-	-	-	-	4	1
Peacock	3	0	0	1	1	0	-	-	-	-	-	3	1
Rockstar	5	4	4	2	2	2	-	-	-	-	-	7	2
Stealers Wheel	9	7	7	3	0	0	-	-	-	-	-	14	3
Thompson Twins	1	1	3	5	4	3	-	-	-	-	-	5	3
Ginger	1	1	0	0	0	-	-	-	-	-	-	1	1
Stealth	1	0	0	0	0	-	-	-	-	-	-	1	1

**Table 2 animals-08-00014-t002:** Colony characteristics: years in program, total cats, age (* <12 weeks of age), sex, and sterilization status.

Colony	Colony Totals		Age		Sex			Sterilized		
Name	Years	Cats	Adult	Kitten *	M	F	Unknown	In program	Previously	Total
James Gang	10	25	25	0	21	4	0	18	3	21
Eleanor Rigby	9	18	13	5	5	10	3	17	0	17
Frontier	8	10	10	0	7	3	0	10	0	10
Garage Band	8	9	9	0	8	0	1	8	0	8
Martino Awesome	8	7	5	2	1	3	3	6	1	7
Jose & The Pussycats	7	13	13	0	3	8	2	10	2	12
Little Sister	7	11	8	3	7	1	3	11	0	11
Marta Volta	7	7	7	0	5	0	2	5	1	6
Mother	7	6	2	4	0	1	5	6	0	6
Ricky Martino	7	6	6	0	5	1	0	6	0	6
V	7	39	37	2	18	14	7	21	12	33
Bonita	6	3	3	0	1	2	0	3	0	3
La Vida Lydia	5	2	2	0	1	0	1	2	0	2
Major Tomcat	5	4	4	0	0	4	0	4	0	4
Peacock	5	4	4	0	3	1	0	1	3	4
Rockstar	5	8	8	0	6	2	0	5	3	8
Stealers Wheel	5	16	13	3	6	4	6	15	0	15
Thompson Twins	5	5	5	0	0	2	3	5	0	5
Ginger	4	1	1	0	0	0	1	1	0	1
Stealth	4	1	1	0	1	0	0	1	0	1
Totals	-	195	176	19	98	60	37	155	25	180
Proportion (%)	-	-	90.3	9.7	50.3	30.8	18.9	79.5	12.8	92.3

**Table 3 animals-08-00014-t003:** Colony size at entry and at calendar year-end.

Colony	Colony Population (Cats)										
Name	Entry	2007	2008	2009	2010	2011	2012	2013	2014	2015	2016
James Gang	3	1	1	3	4	6	5	6	5	6	5
Eleanor Rigby	4	-	4	8	7	7	4	2	1	0	0
Frontier	8	-	-	8	7	7	6	2	2	2	0
Garage Band	1	-	-	0	0	1	0	0	0	1	1
Martino Awesome	2	-	-	0	5	5	0	0	0	0	1
Jose & The Pussycats	3	-	-	-	3	3	8	6	6	5	5
Little Sister	4	-	-	-	0	3	6	6	6	6	6
Marta Volta	3	-	-	-	2	2	0	0	3	3	3
Mother	6	-	-	-	2	3	3	1	1	1	1
Ricky Martino	6	-	-	-	6	6	6	6	6	0	0
V	7	-	-	-	7	15	33	15	15	14	12
Bonita	3	-	-	-	-	3	0	0	0	0	0
La Vida Lydia	1	-	-	-	-	-	1	2	2	1	1
Major Tomcat	4	-	-	-	-	-	4	4	4	4	4
Peacock	3	-	-	-	-	-	0	0	1	1	0
Rockstar	5	-	-	-	-	-	4	4	2	2	2
Stealers Wheel	9	-	-	-	-	-	7	7	3	0	0
Thompson Twins	1	-	-	-	-	-	1	3	5	4	3
Ginger	1	-	-	-	-	-	-	1	0	0	0
Stealth	1	-	-	-	-	-	-	0	0	0	0
Totals	75	1	5	19	43	61	88	65	62	50	44

**Table 4 animals-08-00014-t004:** Colony size at entry, peak, and end, and percentage change at end from entry size and peak size.

Colony	Colony Population (Cats)			Population Change (%)	
Name	Entry	Peak	End (2016)	From Entry	From Peak
James Gang	3	10	5	67	−50
Eleanor Rigby	4	16	0	−100	−100
Frontier	8	10	0	−100	−100
Garage Band	1	4	1	0	−75
Martino Awesome	2	5	1	−50	−80
Jose & The Pussycats	3	11	5	67	−55
Little Sister	4	6	6	50	0
Marta Volta	3	3	3	0	0
Mother	6	6	1	−83	−83
Ricky Martino	6	6	0	−100	−100
V	7	35	12	71	−66
Bonita	3	3	0	−100	−100
La Vida Lydia	1	2	1	0	−50
Major Tomcat	4	4	4	0	0
Peacock	3	3	0	−100	−100
Rockstar	5	7	2	−60	−71
Stealers Wheel	9	14	0	−100	−100
Thompson Twins	1	5	3	200	−40
Ginger	1	1	0	−100	−100
Stealth	1	1	0	−100	−100
Totals	75	152	44	N/A	N/A

**Table 5 animals-08-00014-t005:** Descriptive statistics for colonies, grouped by the number of years tracked. Mean, SD, and median are calculated using “entry” values as first-year values.

Group	No. of Colonies	Years Tracked	Cats per Colony			Total Cats Across Colonies			Population Change (%)	
			Mean	SD	Median	Entry	Peak	End (2016)	From Entry	From Peak
A	1	10	4.4	1.56	5	3	10	5	67	−50
B	1	9	3.7	2.94	4	4	16	0	−100	−100
C	3	8	2.1	2.59	1	11	19	2	−82	−89
D	6	7	5.8	5.82	5.5	29	67	27	−7	−60
E	1	6	0.5	1.12	0	3	3	0	−100	−100
F	6	5	2.7	2.08	2.5	23	35	10	−57	−71
G	2	4	0.3	0.43	0	2	2	0	−100	−100
								Mean	−54	−82
								SD	58.4	19.5
								Median	−82	−89

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
