# Peer review of "A Case Study in Citizen Science: The Effectiveness of a Trap-Neuter-Return Program in a Chicago Neighborhood"

_animals, 2018, doi:10.3390/ani8010014_

Round 1

Reviewer 1 Report

There is nothing scientifically new here, as the ability of long-term TNR to reduce cat population numbers is well established. However, it's worth publishing this work to provide additional data to support the establishment of TNR efforts and to inform broader efforts to create more accurate models of mitigation efforts. A discussion of how these data align or could inform published population models would be of more interest to the field. 

The Methods should be written in a more stringently scientific manner for publication in a scientific journal. In particular, the use of Ms Smetkowski's activities as a methodology is unacceptable. What interventions were applied within what parameters (space and time) and how were the impacts measured? How was the data collected and maintained? There is no description of any statistical analyses provided.  Were any applied?  If not, there should be.  Are the decreases statistically significant?  What is the potential error rate? There is 10 years worth of data in some cases - why not include simple regression analyses to calculate slopes when possible? A statistician should be consulted as to how the data should be analyzed and the findings interpreted.

It should be made clearer throughout, that the population impacts of the programs in Chicago are not entirely due to sterilization because a significant portion have been adopted into homes. It's great to highlight the progress being made in Chicago.

Author Response

Reviewer 2

There is nothing scientifically new here, as the ability of long-term TNR to reduce cat population numbers is well established. However, it's worth publishing this work to provide additional data to support the establishment of TNR efforts and to inform broader efforts to create more accurate models of mitigation efforts. A discussion of how these data align or could inform published population models would be of more interest to the field. 

Thank you for your insightful comments. Our responses are below, in blue type, and were integrated into the manuscript to the extent practicable.

The Methods should be written in a more stringently scientific manner for publication in a scientific journal. In particular, the use of Ms Smetkowski's activities as a methodology is unacceptable. What interventions were applied within what parameters (space and time) and how were the impacts measured?

Details regarding cat description and location were recorded on paper as each cat was trapped, and stored, along with a photo (when available) in a folder designated for the particular colony. Records were updated with additional information (e.g., sex) and transferred to a spreadsheet once each cat was returned to the appropriate colony or transferred to a sponsor organization.

Smetkowski collected census information for each colony on an annual basis, at minimum, through a combination of visual observation (year-round) and informal interviews with colony caretakers (in December of each year). However, it was not unusual for her to track changes in colony membership more frequently. Smetkowski regularly jogged the streets of her neighborhood, past many of the colony sites, and often stopped to check on the cats and their caretakers. She also placed calls to caretakers for updates when she failed to encounter them for more than a few months on her trips through the neighborhood. In addition, Smetkowski would collect census information when colony caretakers contacted her for assistance.

In terms of “impacts,” Smetkowski was concerned only with tracking outcomes for each cat and colony (as described in the text and the five tables uploaded with our revised manuscript). We provide shelter intake data only to provide some sense of the larger context (i.e., the impact of intensive targeted TNR efforts across the Chicago area).

We have added Data Collection and Study Limitations sections to the manuscript.

How was the data collected and maintained?

Details regarding cat description and location were recorded on paper as each cat was trapped, and stored, along with a photo (when available) in a folder designated for the particular colony. Records were updated with additional information (e.g., sex) and transferred to a spreadsheet once each cat was returned to the appropriate colony or transferred to a sponsor organization.

Smetkowski collected census information for each colony on an annual basis, at minimum, through a combination of visual observation (year-round) and informal interviews with colony caretakers (in December of each year). However, it was not unusual for her to track changes in colony membership more frequently. Smetkowski regularly jogged the streets of her neighborhood, past many of the colony sites, and often stopped to check on the cats and their caretakers. She also placed calls to caretakers for updates when she failed to encounter them for more than a few months on her trips through the neighborhood. In addition, Smetkowski would collect census information when colony caretakers contacted her for assistance.

There is no description of any statistical analyses provided.  Were any applied?  If not, there should be.  Are the decreases statistically significant?  What is the potential error rate? There is 10 years worth of data in some cases - why not include simple regression analyses to calculate slopes when possible? A statistician should be consulted as to how the data should be analyzed and the findings interpreted.

Given the relatively small sample size involved, the varying sample sizes for each colony, and the considerable variation in colony populations (as shown in Tables 1 and 3), we worry that linear regression analysis would hinder rather than help one’s understanding of the data. We have consulted with two colleagues familiar with both TNR programs and statistical analysis, both of whom have suggested that we simply include some basic descriptive statistics (Table 5).

It should be made clearer throughout, that the population impacts of the programs in Chicago are not entirely due to sterilization because a significant portion have been adopted into homes. It's great to highlight the progress being made in Chicago.

This is an important point, as we note in Lines 332-334: “The adoption of sociable cats and socializable kittens, commonly considered an essential part of TNR standard operating procedure [7,8,18], contributed significantly to the reductions in population size realized by the CIMY program.”  Adoption also is mentioned on the following lines: 64, 88, 101, 107, 122, 154, 224, 229, and 292-299.

A discussion of how these data align or could inform published population models would be of more interest to the field.

Added to the Discussion, lines 322-325, noting the alignment of CIMY program results and the population modeling of Miller et al. (2014).

Again, thank you for your insightful comments.

Reviewer 2 Report

This paper is a well-written observational study and analysis of changes in free-roaming cat populations in managed cat colonies as well as of changes in intake rates in the Chicago animal shelter system. The paper makes two nice contributions to the literature, first in demonstrating the potential of citizen scientists in informing the management of feral cat colonies, and second in the potential effects of intense TNR programs to reduce feral cat populations.

The authors of the paper should provide additional information to clarify the citizen scientist’s process of data collection, and provide comparison data to determine if the changes in population numbers would be greater than expected in similar programs that did not enact large-scale TNR programs. To clarify, given that ASPCA and others have reported general reductions in shelter intake numbers, how much of the decrease in intake presented in the paper is to be expected given normal attrition, death, adoption, changes in community spay/neuter practices of pets, etc. Does the decrease in intake reported differ than what we would expect without a TNR program. Some sort of financial analysis or feasibility of the given program for other interested citizen scientists would be helpful.

The results section is a bit long and given that there are only percentages and no inferential statistics reported, perhaps some tables or graphs would present this information more efficiently. Specific comments/details below.

Line 78 – please clarify the citizen scientist’s methods for tracking cat populations. How, how often, when, where? How long was a cat required to be absent before they were considered “disappeared”?

Line 87 – how effective was microchipping for tracking outcomes? Did any of the microchipped cats end up at the shelter or veterinary hospitals? Did she routinely scan for microchips when assessing colonies?

Lines 101-102 – again, need more detail on criteria for disappearances

Lines 122-134 This section of the results is a bit hard to follow. Perhaps a table would be a more effective way to present this information. E.g., are the cats who were trapped “collectively” as part of the initial activities different from the other 120? If the spikes are occurring 3-4 years after entry, how reliable is the data?

Line 172 – Were any of the disappeared cats eventually found as DOA in the shelter?

Data from other sources: Again, I wonder if a table would be more effective in presenting this data (and more efficient as well).

Line 208 – Does the intake number exclude cats who went into the TNR program? To bring in my comment from before, does this decrease differ statistically from what would be expected if there wasn’t a TNR program? (See Johnson, 2014; ASPCA stats)

Author Response

Reviewer 1

This paper is a well-written observational study and analysis of changes in free-roaming cat populations in managed cat colonies as well as of changes in intake rates in the Chicago animal shelter system. The paper makes two nice contributions to the literature, first in demonstrating the potential of citizen scientists in informing the management of feral cat colonies, and second in the potential effects of intense TNR programs to reduce feral cat populations.

Thank you for your insightful comments. Our responses are below, in blue type, and have been integrated into the manuscript to the extent practicable.

The authors of the paper should provide additional information to clarify the citizen scientist’s process of data collection, and provide comparison data to determine if the changes in population numbers would be greater than expected in similar programs that did not enact large-scale TNR programs. To clarify, given that ASPCA and others have reported general reductions in shelter intake numbers, how much of the decrease in intake presented in the paper is to be expected given normal attrition, death, adoption, changes in community spay/neuter practices of pets, etc.

Data collection: Details regarding cat description and location were recorded on paper as each cat was trapped, and stored, along with a photo (when available) in a folder designated for the particular colony. Records were updated with additional information (e.g., sex) and transferred to a spreadsheet once each cat was returned to the appropriate colony or transferred to a sponsor organization.

Smetkowski collected census information for each colony on an annual basis, at minimum, through a combination of visual observation (year-round) and informal interviews with colony caretakers (in December of each year). However, it was not unusual for her to track changes in colony membership more frequently. Smetkowski regularly jogged the streets of her neighborhood, past many of the colony sites, and often stopped to check on the cats and their caretakers. She also placed calls to caretakers for updates when she failed to encounter them for more than a few months on her trips through the neighborhood. In addition, Smetkowski would collect census information when colony caretakers contacted her for assistance.

Impact on shelter intake: We don’t feel comfortable drawing direct causal links between the CIMY program and the observed changes in feline intake at local shelters. However, as noted in Sections 3.2.1 and 3.2.2, there is compelling evidence suggesting that TNR efforts in the Chicago area—of which CIMY was a small part—were likely a key factor in the intake reductions (26% from 2010 to 2015) observed. By way of comparison, it is estimated by the American Society for the Prevention of Cruelty to Animals that total feline intake at U.S. shelters decreased from 3.3 million to 3.2 million from 2011 to 2016, a decline of only 3%.

We have added Data Collection and Study Limitations sections to the manuscript.

Does the decrease in intake reported differ than what we would expect without a TNR program.

Our focus is on the reduction of cats via the CIMY project; the shelter intake data is provided only for additional context. For this reason, we don’t feel comfortable providing a statistical analysis of TNR’s impact on shelter intake (though others have done so elsewhere (Edinboro et al. 2016)).

Some sort of financial analysis or feasibility of the given program for other interested citizen scientists would be helpful.

Although we don’t have information sufficient to attempt a financial analysis, we have added the cost of the spay-neuter packages to the text.

Smetkowski had her TNR program surgeries done at either the spay-neuter clinic operated by PAWS or Tree House. The cost at PAWS was $26 and the cost at Tree House was $38 per cat. The feral cat package at both facilities included sterilization, rabies and FVRCP vaccines, ear-tip, flea/ear mite treatment, and microchip (technically, the microchip was optional and $6 of the cost at PAWS and $8 at Tree House, but was required to comply with the MCFC).

The results section is a bit long and given that there are only percentages and no inferential statistics reported, perhaps some tables or graphs would present this information more efficiently. Specific comments/details below.

Four tables were uploaded with the manuscript; we worry, now, that perhaps something went wrong during the process of uploading them. In any event, we have reformatted the tables (including a fifth) for clarity and moved them from Appendix A into the text.

Line 78 – please clarify the citizen scientist’s methods for tracking cat populations. How, how often, when, where? How long was a cat required to be absent before they were considered “disappeared”?

A more detailed description of her methods is included above. In terms of “disappearances,” this refers to the failure to observe a particular cat without knowledge of his/her whereabouts, for six months or more.

Line 87 – how effective was microchipping for tracking outcomes? Did any of the microchipped cats end up at the shelter or veterinary hospitals? Did she routinely scan for microchips when assessing colonies?

Smetkowski did not possess a microchip scanner, so she did not track outcomes for the cats via microchip. However, she was contacted by various entities about eight different cats who were microchipped as part of the CIMY program.

3 at CACC (2 adopted, 1 returned)

2 at Animal Welfare League (1 returned, 1 adopted)

1 at MedVet emergency clinic (euthanized after being hit by a car)

2 at private veterinary practices (both adopted)

Lines 101-102 – again, need more detail on criteria for disappearances

Please see our response above.

Lines 122-134 This section of the results is a bit hard to follow. Perhaps a table would be a more effective way to present this information. E.g., are the cats who were trapped “collectively” as part of the initial activities different from the other 120? If the spikes are occurring 3-4 years after entry, how reliable is the data?

Please see our previous response concerning the four tables (now five) that were uploaded with the manuscript. We have no reason to doubt the reliability of the data. Rather, what appear to be spikes in the size of some colonies are attributed to the fact that cats were counted not as they were first observed by a caretaker but rather as they were trapped or, at least observed by Smetkowski herself (Lines 162-164). Her method of enrolling cats into the program likely prevented overstating of colony sizes by colony caretakers, via double-counting, as described by Centonze and Levy (2002).

Line 172 – Were any of the disappeared cats eventually found as DOA in the shelter?

Data from other sources: Again, I wonder if a table would be more effective in presenting this data (and more efficient as well).

Please see our response above.

Line 208 – Does the intake number exclude cats who went into the TNR program? To bring in my comment from before, does this decrease differ statistically from what would be expected if there wasn’t a TNR program? (See Johnson, 2014; ASPCA stats)

Again, we don’t feel comfortable providing a statistical analysis of TNR’s effect on shelter intake. However, we do find compelling evidence to suggest that TNR efforts (of which CIMY was one small part) played a critical role in reducing feline intake (Lines 290-297):

As was observed in Alachua County, Florida [55], it is likely that intense TNR activity in the city of Chicago contributed significantly to the reductions in feline intake and euthanasia at CACC. Intake was likely lowered because of two interrelated components of TNR: sterilization and adoption. Sterilization likely reduced intake due to fewer kitten births and a reduction in the number of impoundments initiated as a result of resident complaints concerning nuisance feline behaviors commonly associated with mating. Adoption allowed for the removal of sociable cats and kittens directly from TNR programs or after transfer to local rescue organizations, thus eliminating the need for admission to the city shelter.

In addition (Lines 311-315):

The influence of confounding variables (e.g., modifications to shelter polices, animal control policies, or changes in community attitudes) on the apparent associations between TNR and reductions in feline intake and euthanasia at CACC cannot be dismissed entirely, although no such factors were identified as part of this investigation.

Moreover:

Cats admitted to CACC with a tipped ear were returned to their respective colonies whenever possible. Such cats were counted as part of shelter intake. CACC was not a sponsor organization and did not conduct its own TNR program.

Again, thank you for your insightful comments.

Round 2

Reviewer 1 Report

I think authors did an adequate job responding to the comments and that the revised manuscript should be published as is.

Author Response

Thank you.

Reviewer 2 Report

The authors have done an excellent job at addressing my concerns and clarifying the limitations of the data. I see only one problem, which is that Table 4 -- the header says Population change, but what is presented is not the change but the percent in relation to the reference (entry or peak). Either the header or the data should be corrected, similar to Table 5. 

Author Response

Modified Table 4 per your comments. Thank you for catching the discrepancy.